# Feasibility of gel-like radiopaque embolic material using gelatin sponge and contrast agent for tract embolization after percutaneous treatment

Takehito Nota[�}], Ken Kageyama[iD][☾]*, Akira Yamamoto[☾], Atsushi Jogo[‡], Etsuji Sohgawa[‡], Hiroki Yonezawa[‡], Kazuki Murai[iD][‡], Satoyuki Ogawa[‡], Yukio Miki[‡]

Department of Diagnostic and Interventional Radiology, Graduate School of Medicine, Osaka Metropolitan University, Osaka, Japan

☾ These authors contributed equally to this work.
‡ AJ, ES, HY, KM, SO and YM also contributed equally to this work.
* kageyama@omu.ac.jp

## Abstract

### Objectives

Tract embolization has been performed to prevent bleeding after trans-organ puncture. This study evaluated clinical outcomes of tract embolization using a gel-like radiopaque material comprising two sheets of gelatin sponge and 3 mL of contrast agent, and experimentally confirmed its viscosity and hemostatic efficacy.

### Methods

Three study phases were planned. In a clinical setting, 57 consecutive patients who underwent tract embolization after transhepatic puncture were retrospectively analyzed. Clinical success was evaluated as absence of bleeding complications for 30 days after the procedure. In a basic experiment, viscosity of the material was analyzed. In an animal experiment, rabbit kidney puncture site was embolized via a 7-Fr sheath using this material, coils, or N-butyl-2-cyanoacrylate glue or received no embolization while removing the sheath. Amounts of tract bleeding were measured for 1 min and compared between groups.

### Results

Embolization was successfully completed in all clinical cases. No postoperative bleeding requiring intervention was encountered. The basic experiment revealed the material was highly viscous. In the animal experiment, mean weights of bleeding in the control, gel-like embolic material, coil, and N-butyl-2-cyanoacrylate glue groups were 1.04±0.32 g, 0.080 ±0.056 g, 0.20±0.17 g and 0.11±0.10 g, respectively. No significant differences were seen among embolization groups, while the control group showed significantly more bleeding than any embolization group.

**Data Availability Statement:** All relevant data are within the manuscript and its Supporting information files.

**Funding:** The authors received no specific funding for this work.

**Competing interests:** The authors have declared that no competing interests exist.

## Conclusion

Tract embolization with this gel-like radiopaque embolic material appears safe and feasible.

## Advances in knowledge

Tract embolization using this embolic material with two sheets of gelatin sponge and 3 mL of contrast agent offers a safe, feasible, and economical procedure after trans-organ puncture, because the material offers the following characteristics: visibility under X-ray; viscosity facilitating retention in the tract; ability to allow repeated puncture via the same route; and low cost.

## Introduction

Percutaneous transhepatic puncture has frequently been performed for percutaneous biopsy, percutaneous transportal embolization (PTPE), percutaneous transhepatic biliary drainage (PTBD), and percutaneous transhepatic sclerotherapy (PTS) for esophagogastric varices [1–4]. These procedures represent minimally invasive diagnostic and treatment alternatives to invasive surgical approaches [1, 2, 5].

However, these puncture procedures carry a risk of causing tract bleeding after removal of the puncture needle or sheath [6, 7]. Arterial and portal bleeding from the puncture tract are critical complications after transhepatic puncture [8, 9]. A previous report revealed that hemorrhage after hepatic puncture sometimes leads to death [10]. Two previous reports demonstrated hemorrhage following percutaneous transhepatic portal interventional therapy in about 10% of cases [11, 12]. Some bleeding cases showed asymptomatic intraperitoneal hemorrhage, which may delay detection of the hemorrhage [12]. Life-threatening hemorrhage can require interventions such as transfusion therapy, transarterial embolization therapy, and surgery [10–13].

Tract embolization is an essential procedure to control bleeding. Previous studies have reported this procedure using various materials, including cylindrical gelatin sponges [14], N-butyl-2-cyanoacrylate (NBCA) glue [13], coils [15, 16], and vascular plugs [17, 18]. Although the efficacies of each embolic material have been reported individually, comparative evaluations have been lacking in clinical, prospective settings. The hemostatic effects of embolic materials can be evaluated in animal experiments prior to clinical application.

To prevent life-threatening hemorrhage, embolic materials are required for tract embolization. We have been routinely using embolic materials comprising gelatin sponge and contrast agent for tract embolization after transhepatic procedures in clinical practice for over a decade. However, we have not yet evaluated our own embolic material. The purpose of this study was to evaluate the efficacy of tract embolization after percutaneous puncture using a gel-like embolic material comprising two sheets of gelatin sponge and 3 mL of contrast agent. In clinical practice, we retrospectively evaluated the hemostatic efficacy of this material after the procedure. In a basic experiment, we investigated the viscosity of this embolic material by comparing different amounts of contrast agent. In an animal experiment, we compared the hemostatic efficacy of this material with that of coils and NBCA glue in rabbit kidney.

## Materials and methods

### Clinical practice

**Patients and characteristics.** All procedures in this single-center retrospective study were approved by the institutional review board (approval no. 2020059), and the requirement to

obtain informed consent was waived because of the retrospective design. Three radiologists, each with more than 10 years of experience in diagnostic and interventional radiology, retrospectively analyzed 69 consecutive patients who underwent percutaneous transhepatic endovascular therapies and embolization of transhepatic puncture tracts with the embolic material described below at our facility between January 2014 and September 2019. All transhepatic punctures were performed under ultrasonographic guidance (LOGIQ S7 Expert machine equipped with a C1-5-D convex probe; GE Healthcare, Tokyo, Japan) by interventional radiologists with over 10 years of experience. In all 69 patients, transhepatic puncture tracts were routinely embolized at our facility when endovascular treatment of the portal vein was performed via a percutaneous transhepatic route. Dynamic contrast-enhanced computed tomography (CT) was routinely performed both before the transhepatic puncture and within one month after the procedure. Non-enhanced CT was also performed just after the procedure. Twelve patients were excluded because they lacked contrast-enhanced CT images from within 30 days after the procedure. Characteristics of the 12 excluded patients are summarized in S1 Table. Consequently, 57 patients were investigated in this study (S1 Fig). Included patients and their characteristics are summarized in Table 1.

**Making the embolic material.** Two sheets of absorbable gelatin sponge (Spongel, 2.5 cm × 5 cm × 1 cm; LTL Pharma, Tokyo, Japan) were roughly cut into 5 mm × 5 mm squares with a knife. All these squares were loaded into a 5-mL syringe, then cut finely with 13-cm-long dressing scissors inside the syringe barrel and packed in the top of the barrel to prevent them from overflowing. After compressing the chunks of cut sponge to remove air, the syringe was connected via a three-way stopcock to a syringe containing 3 mL of contrast agent (Omnipaque 300; GE Health Care, Tokyo, Japan). We then pumped the syringe five times to create a gel-like radiopaque embolic material (S1 Video). All processes were completed within 10 minutes.

**Procedure.** The syringe filled with gel-like radiopaque embolic material was connected to the side arm of a sheath inserted into the liver. Three milliliters of the embolic material were injected into the intrahepatic tract from the portal vein orifice at the liver surface using a finger to seal the check valve of the sheath, while the sheath was carefully retracted under X-ray fluoroscopy, paying attention to non-target embolization of the portal veins and hepatic veins (S2 Video).

**Follow-up.** The primary endpoints in this study were the technical and clinical success of hepatic tract embolization. Technical success was defined as visualization of this embolic material under X-ray fluoroscopy and no migration of stagnant embolic material during X-ray fluoroscopy. Non-enhanced CT was performed immediately after the procedure. Clinical success was defined as a lack of bleeding complications requiring blood transfusion, transarterial embolization or surgical intervention within 30 days after the procedure. Patients were continuously monitored for vital signs such as heart rate and blood pressure for 24 h after the procedure. From the day after the procedure (Day 1), signs were measured a few times a day during hospitalization. Blood tests, including blood cell counts, hepatic function, renal function, and coagulation function, were examined on Day 1. Blood tests were performed every two days during hospitalization. The clinical course was documented for each patient for one month after the procedure. The puncture tract and portal veins were evaluated on contrast-enhanced CT within 30 days after the procedure. Interventional radiologists who performed the procedure evaluated technical success just after the procedure and clinical success during the follow-up period. After discharge, patients were followed-up by interventional radiologists at our facility. Post-procedural complications were graded using Common Terminology Criteria for Adverse Events (CTCAE) version 5.0.

**Table 1. Characteristics and clinical results of patients who underwent hepatic tract embolization.**

| Characteristic | Value |
|---|---|
| **No. of patients (M/F)** | 57 (36/21) |
| **Age, years, mean ± SD** | 66 ± 9.3 |
| **Pre-existing diseases** | |
| Cholangiocellular carcinoma | 18 |
| Liver metastasis | 11 |
| Gastric varix | 8 |
| Esophageal varix | 8 |
| Rectal varix | 4 |
| Gallbladder cancer | 2 |
| Portal vein thrombosis | 2 |
| Hepatocellular carcinoma | 2 |
| Other | 2 |
| **Child-Pugh classification** | |
| A / B / C | 33 / 23 / 1 |
| **Treatment** | |
| PTPE | 32* |
| PTS | 22 |
| Portal stenting | 1 |
| Other | 2 |
| **Portal vein for percutaneous access route** | |
| Right portal vein | 3 |
| Anterior branch of right portal vein | 22 |
| Posterior branch of right portal vein | 15 |
| Left portal vein | 17 |
| **Sheath size** | |
| 4-Fr | 42 |
| 5-Fr | 7 |
| 6-Fr | 7 |
| 7-Fr | 1 |
| **Platelet count, ×10$^4$/μL, mean ± SD** | 16.6 ± 9.5 |
| **Hospital stay, days, mean ± SD (range)** | 9.4 ± 7.5 (3–51) |
| **Postoperative days to CT evaluation, days, mean ± SD (range)** | 11.3 ± 8.3 (1–30) |
| **Adverse Events** | |
| [†]**Postoperative hemorrhage** | |
| No complication | 55 (96.5%) |
| Grade 1 | 2 (3.5%) |
| [†]**Portal vein thrombosis** | |
| No complication | 55 (96.5%) |
| Grade 3 | (3.5%) |

PTPE, percutaneous transportal embolization; PTS, percutaneous transhepatic sclerotherapy; SD, standard deviation.

\* Ipsilateral puncture in 29 cases; contralateral puncture in 4 cases.

[†] As graded by Common Terminology Criteria for Adverse Events (CTCAE) version 5.0.

## Basic experiments

**Viscosity.**   To reveal the viscosity of embolic materials, a comparative study was conducted. The embolic materials were made in the same way described above. The amount of contrast

agent was changed when the embolic materials were generated. One, two, three, four, or five milliliters of contrast agent was mixed with two sheets of absorbable gelatin sponge. The dynamic viscosity of these embolic materials was determined using a Brookfield viscometer (DV2TCP; Brookfield Engineering Laboratories; Middleboro, MA, USA) equipped with a cone spindle (CPA-51Z; Brookfield Engineering Laboratories). Viscosity of the five different embolic materials was measured 60 times in 10 min at 25˚C. Measurements were performed at 2.4 revolutions per minute (RPM). The shear rate was 9.216 s$^{-1}$. In addition, viscosity was also measured for normal saline, iodinated contrast agent, ethyl ester of iodinated poppy-seed oil fatty acid (Lipiodol; Guerbet Japan, Tokyo, Japan), and embolic material with two sheets of gelatin sponge and 3 mL of normal saline water as reference data. Viscosity was measured for the embolic material with two sheets of gelatin sponge and 3 mL of normal saline water using the same RPM setting. The other three materials were measured at different RPM settings to confirm Newtonian behavior. We assessed the ability for these embolic materials to be passed through the 4-Fr system sheath (Medikit Super Sheath; Medikit, Tokyo, Japan) using hand pressure.

## Animal experiments

**Ethics and animals.** All experiments were performed at our institution under a protocol approved by the animal care committee. Animals were four 12-week-old Japanese white rabbits weighing 2–2.5 kg. After evaluating acute bleeding associated with the procedure, euthanasia was humanely performed using intravenous administration of high-dose ketamine and xylazine under adequate anesthesia, as described below. Kidney was chosen as the organ to test because bleeding was easier to create than in the liver in a preliminary experiment.

**Procedure for animal experiment.** Each rabbit underwent induction of anesthesia by intramuscular injection of 1 mL of a mixture of ketamine 40 mg (0.8 mL) and xylazine 4 mg (0.2 mL) into a thigh muscle. Intravenous injection of 0.1 mL of the same mixture was also intermittently performed via an auricular vein for maintenance of anesthesia. Under general anesthesia, the retroperitoneal cavity was surgically opened to expose bilateral kidneys. Prior to puncture and tract embolization, 3 mL of contrast agent was administered intravenously in the rabbit. Contrast-enhanced kidneys were identified under X-ray fluoroscopy. The renal parenchyma was punctured using a 7-Fr system sheath (Medikit Super Sheath; Medikit). A total of six punctures per animal was performed, with each kidney punctured three times. Each puncture was performed radially in the direction of the renal hilum under X-ray fluoroscopy. During sheath withdrawal, tract embolization was performed with each of the following embolic materials: 1 mL of gel-like embolic material; coils (Tornado Embolization Coil, 4/2 mm; Cook, Bloomington, IN, USA); and NBCA glue (Histoacryl; B. Braun, Melsungen, Germany). NBCA glue was a mixture of 0.5 mL of NBCA and 2 mL of lipiodol. NBCA glue was injected to the extent that it did not adhere to the sheath. Each embolic material was used for tract embolization in one rabbit. The control group did not undergo tract embolization.

**Follow-up evaluation.** Visibility of the three embolic materials was confirmed during the procedure. Gauze was prepared to absorb blood from the puncture site for 1 min after sheath removal. Each gauze was then weighed, and the amount of bleeding was recorded for each group. After the procedure, rabbits were humanely sacrificed.

**Pathological analysis.** Kidneys filled with gel-like embolic material in the tract were extracted immediately after sacrifice, then fixed in 10% buffered formalin. After fixation, serial sections were made to identify the lesions. Representative sections from each lesion and the interface between lesion and normal parenchyma were selected for paraffin embedding and stained with hematoxylin and eosin (HE). HE-stained slides were examined for the presence of gelatin sponge.

### Statistical analysis

In the basic experiment, statistical comparison of viscosity between the four embolic materials was performed via Student's t test with Bonferroni correction. In the animal experiment, statistical comparisons of the four groups were performed using the Mann-Whitney U test with Bonferroni correction. Statistical analyses were performed using GraphPad Prism version 8.00 (GraphPad Software, La Jolla, CA, USA). Values of $P < 0.0083$ were considered to indicate a significant difference.

## Results

### Clinical practice

Tract embolization was technically successful in all cases (Fig 1A and 1B). This embolic material was visible under X-ray during all procedures. Non-enhanced CT identified no migration of embolic material from the tract and no accumulation of hyperdense fluid surrounding the liver (Fig 1C). The two patients who developed subcutaneous hematomas at the puncture site were classified as Grade 1. Clinical success was achieved in all cases because no signs of bleeding from the transhepatic puncture route were clinically observed during the 1-month follow-up after the procedure. The area of high concentration in the hepatic tract eventually disappeared on follow-up CT in all 57 cases (Fig 1D). Similarly, portal veins were visualized on CT in all 57 cases. Non-target embolization in the portal veins was not apparent on CT in 55 of the 57 cases (Fig 1D). The other two cases showed peripheral portal vein emboli, classified as Grade 3 (Intermediate) on CT (S2 Fig). Anticoagulation using danaparoid sodium or direct-acting oral anticoagulant was started 3 days after the procedure in these two cases, and portal vein emboli subsequently disappeared on follow-up CT. No other complications were observed during follow-up.

### Basic experiment

One milliliter of contrast agent failed to mix with two sheets of absorbable gelatin sponge due to the insufficiently aqueous nature. The other four arms with 2–5 mL of contrast agent showed complete mixing with gelatin sponge. Results for viscosity are shown in Table 2. The arm with 2 mL of contrast agent showed the highest viscosity compared to the other three arms with 3–5 mL of contrast agent (Fig 2). Viscosity differed significantly between each arm (2 mL vs. 3 mL, p < 0.0001; 2 mL vs. 4 mL, p < 0.0001; 2 mL vs. 5 mL, p < 0.0001; 3 mL vs. 4

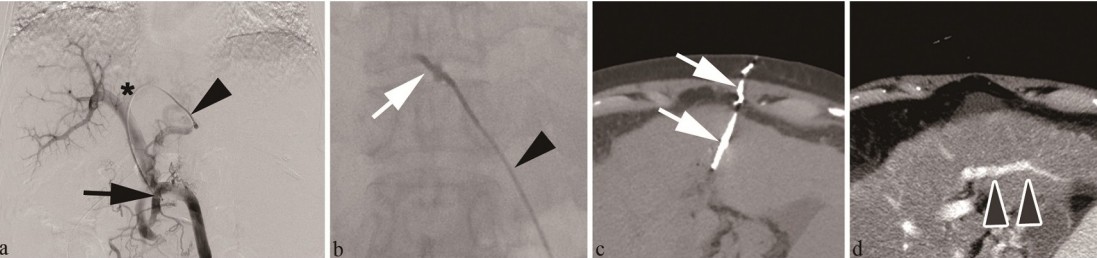

**Fig 1. Hepatic tract embolization after percutaneous transhepatic sclerotherapy (PTS) in a 71-year-old woman with rectal varices.** a) Portal venography obtained during PTS shows the catheter (black arrow) advanced through a 6-Fr sheath (black arrowhead) via the left branch of the portal vein (asterisk) under a transhepatic approach. b) Anterior radiography shows injection of gel-like radiopaque embolic material (white arrow) into the tract through the sheath (black arrowhead). c) Non-contrast-enhanced computed tomography (CT) confirms retention of radiopaque embolic materials (white arrows) in the hepatic tract. d) The contrast effect of the embolic material on contrast-enhanced CT disappeared 3 months after tract embolization. The left anterior portal branches visualized on contrast-enhanced CT (gray arrowhead).

**Table 2. Viscosity of all embolic materials and substances.**

| Embolic materials and substances | Viscosity (mPa·s) |
|---|---|
| Two sheets of gelatin sponge and 2 mL of contrast agent, mean ± SD (range) | 81,908 ± 3568.4 (78,710–95,790) |
| Two sheets of gelatin sponge and 3 mL of contrast agent, mean ± SD (range) | 61,634 ± 4275.2 (50,570–68,700) |
| Two sheets of gelatin sponge and 4 mL of contrast agent, mean ± SD (range) | 33,942 ± 1926.9 (28,480–36,760) |
| Two sheets of gelatin sponge and 5 mL of contrast agent, mean ± SD (range) | 22,309 ± 1177.2 (16,740–23,820) |
| Normal saline water | 2.8 |
| Iodinated contrast agent | 17.1 |
| Ethyl ester of iodinated poppy-seed oil fatty acid | 33.3 |
| Two sheets of gelatin sponge and 3 mL of saline water, mean ± SD (range) | 43,389 ± 3181.0 (38,320–52,640) |

**Note**. SD; standard deviation

mL, $p < 0.0001$; 3 mL vs. 5 mL, $p < 0.0001$; 4 mL vs 5 mL, $p < 0.0001$.). In the arm with 2 mL of contrast agent, the agent proved unable to pass through the 4-Fr sheath, while the others could. Consequently, the arm with 3 mL of contrast agent had the most viscous material that could be injected through the sheath.

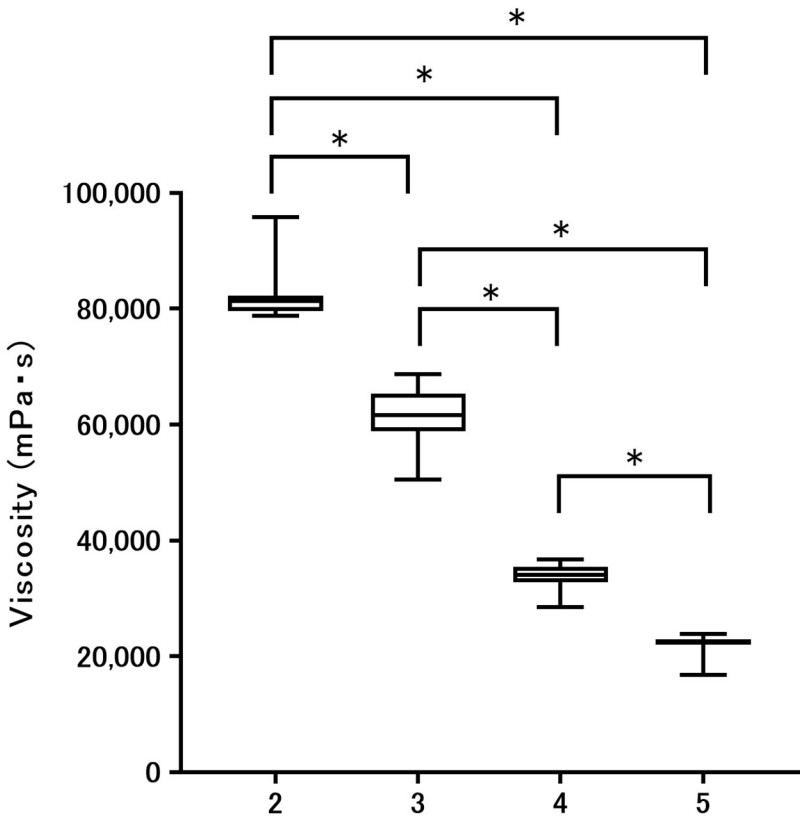

**Fig 2. Viscosity of gel-like embolic material using two sheets of gelatin sponge with contrast agent.** *$p < 0.0083$: 2 mL vs. 3 mL, $p < 0.0001$; 2 mL vs. 4 mL, $p < 0.0001$; 2 mL vs. 5 mL, $p < 0.0001$. 3 mL vs. 4 mL, $p < 0.0001$, 3 mL vs 5 mL, $p < 0.0001$, 4 mL vs 5 mL, $p < 0.0001$.

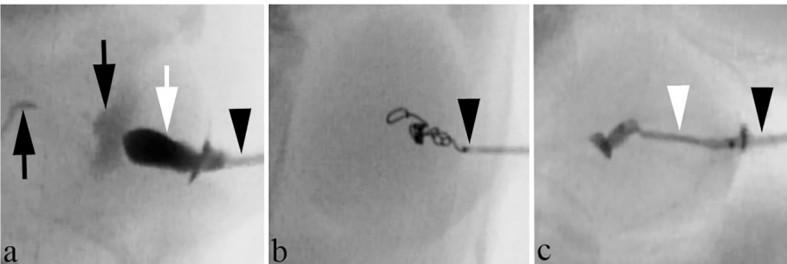

**Fig 3. Tract embolization after renal puncture in a rabbit.** The kidney was intravenously injected with contrast agent to allow operator understanding of the outline of the kidney. a) A 7-Fr sheath (black arrowhead) is inserted into the contrast-enhanced kidney. The renal pelvis and ureter (black arrows) also show contrast enhancement. Gel-like embolic material (white arrow) is injected through the sheath while withdrawing the sheath. b) A coil is inserted in the puncture tract through the 7-Fr sheath (black arrowhead). c) Injection of 0.5 mL of NBCA glue (white arrowhead) through the 7-Fr sheath (black arrowhead) while withdrawing the sheath.

## Animal experiment

All procedures were successfully performed, and animals were humanely sacrificed as scheduled (Fig 3). Three embolic materials were visible during all procedures. Mean weights of bleeding in the control, gel-like embolic material, coil and NBCA groups were 1.04±0.32 g, 0.080±0.056 g, 0.20±0.17 g and 0.11±0.10 g, respectively (Fig 4). Hemorrhage volume was significantly larger in the control group than in the other three groups (vs. gel-like embolic material, p = 0.0007; vs. coil, p = 0.0027; vs. NBCA, p = 0.0013). No significant differences were seen between the embolic agent and the other three groups (gel-like embolic material vs. coil, p = 0.4848; gel-like embolic material vs. NBCA, p = 0.9372; coil vs. NBCA, p = 0.5887).

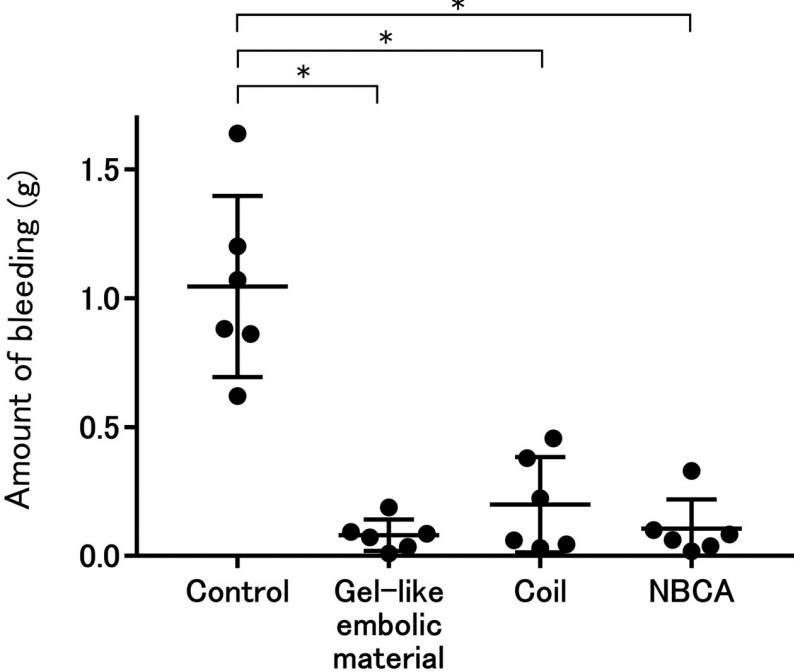

**Fig 4. Comparison of the amount of bleeding after puncture of the rabbit kidney.** *p < 0.0083: control vs. gel-like embolic material, p = 0.0007; control vs. NBCA, p = 0.0027; control vs. coil, p = 0.0013; gel-like embolic material vs NBCA, p = 0.4848; gel-like embolic material vs coil, p = 0.9372; NBCA vs coil, p = 0.5887.

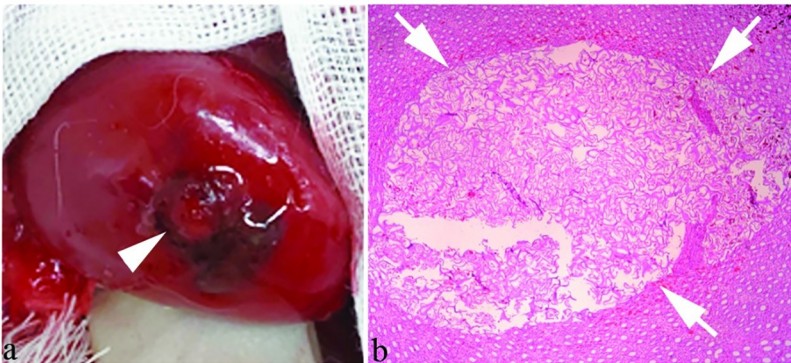

**Fig 5. Rabbit kidney after tract embolization with gel-like embolic material.** a) Macroscopic image shows overflow of gel-like embolic material (white arrowhead) from the puncture site. b) Microscopic image of renal parenchyma shows the punctured tract entirely filled with injected gel-like gelatin sponge particles (white arrows).

Macroscopic imaging showed gel-like embolic material occupying the renal tract (Fig 5A). Microscopic images of renal parenchyma demonstrated the punctured tract entirely filled with gelatin sponge particles, along with a small amount of blood (Fig 5B).

## Discussion

This study demonstrated that gel-like radiopaque embolic material using two sheets of gelatin sponge and 3 mL of contrast agent was efficient for tract embolization of the puncture route. The hepatic puncture route did not show any hemorrhage after the procedure in any clinical cases. In the basic experiment, the gel-like radiopaque embolic material displayed a high degree of viscosity and was able to pass through a 4-Fr sheath by hand injection. In the animal experiment, the gel-like radiopaque embolic material was similar to the other embolic materials in terms of hemostatic efficacy. All three phases of the study indicated that this method is safe and feasible for tract embolization.

Embolic materials need to be visible under X-ray fluoroscopy to allow confirmation of tract embolization. Particularly in transportal procedures, radiopaque materials and devices have been used to ensure tract closure [19, 20]. However, gelatin sponge itself was relatively hard to visualized, appearing almost radiolucent during the embolization procedure under X-ray fluoroscopy [21, 22]. Clinicians cannot confirm tract hemostasis with such radiolucent embolic materials. In the present study, gel-like embolic material using two sheets of gelatin sponge and 3 mL of contrast agent was radiopaque under X-ray fluoroscopy. We could confirm the location of these embolic materials during the procedure.

High viscosity is required as a characteristic of embolic materials during tract embolization to avoid bleeding from the tract as a major complication. The embolic materials should thus be retained in the tract without migration. In tract embolization using gelatin sponge particles with normal saline or gelatin sponge torpedo, whether the sponge was completely retained within the tract remained uncertain under X-ray fluoroscopy. Clinicians have voiced concerns about the displacement of such conventional materials using gelatin sponge from the tract, which might result in tract bleeding [23, 24]. The shape of gelatin sponge particles may affect the likelihood of incomplete tract embolization and delayed bleeding due to their soluble, impermanent nature [13, 22]. A previous report confirmed closure of the transhepatic route by injecting a slurry made with 1 sheet of Gelfoam and 3 mL of contrast media [25]. The volume of gelatin sponge in the previous study was smaller than that in our study. Our gel-like embolic material might have been more viscous material than the material used in the previous

study. We believe that viscous substances such as our gel-like embolic materials are suitable for tract embolization without displacement. This material was observed on CT to occupy and be retained in the tract after the procedure. In the basic experiment, the mixture of two sheets of gelatin sponge and 2 mL of contrast agent showed high viscosity. However, this material proved too viscous to pass through the 4-Fr system sheath. We therefore decided that the mixture of two sheets of gelatin sponge and 3 mL of contrast media offered optimal viscosity as an embolic substance.

The embolic material for tract puncture should allow repeated procedures via the same route. Tract embolization has been performed using various embolic substances, such as coils [15], NBCA glue [13, 16] and vascular plugs [17]. NBCA glue injection carries concerns about non-target embolization and glue detained focally in the tract is difficult to handle [26]. The use of NBCA glue requires considerable technical skill and experience. Furthermore, the procedure must be completed in one operation without stopping the injection on the way to prevent adhesions between the catheter and NBCA. Coils and vascular plugs can also be used as embolic materials to achieve complete embolization. However, the risk of migration and a longer procedure time are disadvantages [13]. Tract bleeding is sometimes difficult to observe with coils and vascular plugs because of metal artifacts on CT. Our embolic material allows slow injection that fills the tract under X-ray fluoroscopy. If non-target lesions are depicted, infusion of the embolic material can be suspended, unlike NBCA and coils. In our methods, contrast enhancement of our gel-like radiopaque embolic material definitely disappeared from the tract on follow-up CT. In one case, follow-up CT revealed a high-concentration area in the tract that disappeared within just 1 day after the procedure. With this tract embolization, repeated percutaneous punctures can be performed via the same route without being disturbed by artifacts of NBCA glue, coils, or plugs.

The costs of each embolic material per procedure in Japan were $4 for gel-like materials, $35 for NBCA, $100–1,000 for coils, and $1,000 for Amplatzer vascular plugs, respectively. Gelatin sponge is thus relatively inexpensive as an embolic material. In our animal experiment, the gel-like embolic material offered hemostatic effects that were statistically comparable with those of the other materials. Assuming that the embolic potential does indeed prove equivalent to that of other materials, we believe that this cost-saving embolic substance will offer benefits in terms of healthcare economics.

This study had several limitations. First, the clinical study involved a retrospective study of a single institution with a small sample size, and did not include a control group. Second, clinical results for this gel-like embolic material were not able to be directly compared with those of other embolic materials. We thus had to confirm the efficacy of this embolic material in an animal experiment. Third, in the animal study, the puncture target was different from that used in clinical practice. As a pilot experiment, we attempted to puncture the rabbit liver and spleen. However, little bleeding was observed from those organs. The kidney has an abundant blood supply and bleeds readily after puncture, and so was considered an appropriate organ for evaluating tract embolization.

In conclusion, tract embolization using this embolic material with two sheets of gelatin sponge and 3 mL of contrast agent is a safe, feasible, and economical procedure after transorgan puncture.

## Supporting information

**S1 Table. Characteristics and clinical results of patients who underwent hepatic tract embolization from 12 excluded patients.**
(DOCX)

**S1 Fig. Flowchart of this study.**
(TIF)

**S2 Fig. A case of portal vein emboli on CT due to non-target embolization.** a) Post-embolization cone-beam CT showed spotty hyper-attenuation in the umbilical portion of the portal vein (white arrowhead) away from the embolized transhepatic tract (white arrow). b) Contrast-enhanced CT three days after the procedure did not visualize the left anterior branch or umbilical portion of the portal vein. c) Follow-up CT showed visualization of the left anterior branch and umbilical portion after anticoagulation therapy for three months.
(TIF)

**S1 Video. The process of making the gel-like embolic material.**
(MP4)

**S2 Video. Hepatic tract embolization after percutaneous transhepatic sclerotherapy (PTS) in a 71-year-old woman with rectal varices.** This video corresponds to Fig 1.
(MP4)

## Acknowledgments

We gratefully acknowledge the work of past and present members of our facility.

## Author Contributions

**Conceptualization:** Akira Yamamoto.

**Data curation:** Takehito Nota, Ken Kageyama.

**Funding acquisition:** Ken Kageyama.

**Investigation:** Takehito Nota, Ken Kageyama, Akira Yamamoto, Atsushi Jogo, Etsuji Sohgawa, Hiroki Yonezawa, Kazuki Murai, Satoyuki Ogawa.

**Methodology:** Ken Kageyama, Akira Yamamoto, Atsushi Jogo, Etsuji Sohgawa, Hiroki Yonezawa, Kazuki Murai, Satoyuki Ogawa.

**Project administration:** Takehito Nota, Ken Kageyama, Akira Yamamoto, Atsushi Jogo, Etsuji Sohgawa, Hiroki Yonezawa, Kazuki Murai, Satoyuki Ogawa.

**Resources:** Takehito Nota, Ken Kageyama, Atsushi Jogo, Etsuji Sohgawa, Hiroki Yonezawa, Kazuki Murai, Satoyuki Ogawa.

**Supervision:** Akira Yamamoto, Yukio Miki.

**Visualization:** Takehito Nota.

**Writing – original draft:** Takehito Nota, Ken Kageyama.

**Writing – review & editing:** Takehito Nota, Ken Kageyama, Akira Yamamoto, Yukio Miki.

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
