## [Decision Letter · Decision Letter 0]

10 Aug 2022

PONE-D-22-16655Feasibility of gel-like radiopaque embolic material using gelatin sponge and contrast agent for tract embolization after percutaneous treatmentPLOS ONE

Dear Dr. Kageyama,

Thank you for submitting your manuscript to PLOS ONE. After careful consideration, we feel that it has merit but does not fully meet PLOS ONE’s publication criteria as it currently stands. Therefore, we invite you to submit a revised version of the manuscript that addresses the points raised during the review process.

Please note that we have only been able to secure a single reviewer to assess your manuscript. We are issuing a decision on your manuscript at this point to prevent further delays in the evaluation of your manuscript. Please be aware that the editor who handles your revised manuscript might find it necessary to invite additional reviewers to assess this work once the revised manuscript is submitted. However, we will aim to proceed on the basis of this single review if possible. 

We look forward to receiving your revised manuscript.

Kind regards,

Vanessa Carels

Staff Editor

PLOS ONE

Journal Requirements:

2. Please provide additional details regarding participant consent. In the Methods section, please ensure that you have specified (1) whether consent was informed and (2) what type you obtained (for instance, written or verbal). If your study included minors, state whether you obtained consent from parents or guardians. If the need for consent was waived by the ethics committee, please include this information.

4. Please include a copy of Table 2 which you refer to in your text on page 17. 

Reviewers' comments:

Reviewer's Responses to Questions

**Comments to the Author**

1. Is the manuscript technically sound, and do the data support the conclusions?

Reviewer #1: Yes

2. Has the statistical analysis been performed appropriately and rigorously? 

Reviewer #1: Yes

3. Have the authors made all data underlying the findings in their manuscript fully available?

Reviewer #1: Yes

4. Is the manuscript presented in an intelligible fashion and written in standard English?

Reviewer #1: Yes

5. Review Comments to the Author

Reviewer #1: PONE-D-22-16655

Feasibility of gel-like radiopaque embolic material using gelatin sponge and contrast

agent for tract embolization after percutaneous treatment

Dear Editors of PLOS ONE,

thank you very much for the opportunity to review manuscript PONE-D-22-16655, entitled " Feasibility of gel-like radiopaque embolic material using gelatin sponge and contrast agent for tract embolization after percutaneous treatment".

In this article the authors present their experiences with a special embolic agent in the context of percutaneous transhepatic procedures.

The originality of content appears satisfying, as I am not aware of any other literature that has examined the use of such an embolic material. The broad approach, including not only the analysis of clinical data, but also experimental methods, makes the study well-founded. The utilized statistical methods are suitable for the evaluated data. The tables and figures are appropriate. In my opinion, publication of this manuscript can be recommended with minor revisions.

Introduction

1. I miss the mention of percutaneous transhepatic biliary interventions, e.g. percutaneous transhepatic biliary drainage (PTBD), in the first paragraph.

2. I consider the sentence “The hemostatic effect of embolic materials requires evaluation in animal experiments.” in line 72 to be too strict and not necessary.

3. In my opinion, the introduction is too lengthy and should be streamlined. For example, the information given in lines 55 – 56 is negligible.

4. Line 57 – 58: “However, these procedures carry a risk of accidentally puncturing structures such as arteries, portal veins, and veins, which would result in haemorrhage”. I find this sentence confusing, since in the setting of the presented procedures the underlying approach is the intentional creation of a transportal access. It would be appropriate if percutaneous biopsies were involved.

Methods

5. Considering the retrospective study setting, I was surprised that non-enhanced CT was performed immediately after the procedure in all of the included cases. Is this regarded the internal standard in your department? Was the imaging of the embolic agent distribution the only indication for this examination? This aspect might be discussed in the context of radiation safety.

6. Was exactly 3 ml of the embolic material always used? I would expect the amount to depend on the length and size of the intrahepatic catheter tract.

7. Line 136 – 138: “Non-enhanced CT performed immediately after the procedure identified no migration of embolic material from the tract and no hyperdense fluid accumulation surrounding the liver.“

This paragraph should rather be moved to the results section.

8. Animal experiment: How many punctures were performed per animal? Based on Figure 4, I would expect that six attempts were made per embolization material? Was only one type of embolic material used in the same animal? You should provide more information on this.

9. Furthermore, the extent of bleeding may depend on the location of the puncture tract. A central puncture at the level of the renal hilus is more likely to lead to a relevant bleeding than a puncture at the upper or lower peripheral organ pole. If I understand correctly, the kidney of the animals were punctured blindly. Thus, there was no control of whether a larger vessel was punctured or not. How could it be ensured that the bleeding severity was comparable for each puncture? That aspect might be also worth mentioning in the limitation section.

10. I don´t quite understand why the kidneys were i.v. contrasted when they were surgically exposed for the puncture anyway.

11. Why was the pathological analysis of the embolized kidneys only carried out with the gel-like embolic material? The cases after coil embolization may be difficult to process as serial sections, but those after NBCA embolization could have been interesting, especially in comparison to the gel-like embolic material.

12. What was the embolic agent of your choice before the described gel-like material was introduced?

Results

13. In the cases of non-target embolization, what was the percutaneous access route? Do the authors believe the access route might potentially influence the occurrence of such complications?

14. The authors describe two cases of subcutaneous hematomas at the puncture site. Did non-enhanced post-procedural CT show no perihepatic fluid accumulation in these cases either?

Discussion

15. I think the discussion is quite lengthy and could be condensed.

16. Since the presented study deals with percutaneous tract embolization after transportal interventions, I think some of the literature cited in line 306 – 311 is not suitable. On the other hand, I miss literature that explicitly deals with the embolization of access tract after percutaneous transportal procedures, such as:

- Embolization of the Transhepatic Tract after Percutaneous Portal Vein Interventions: Single-Centre Retrospective Study Comparing n-butyl Cyanoacrylate Versus Coils by Zi-Han Zhang et al.

- Feasibility of Mynxgrip®-Assisted Percutaneous Transhepatic Portal Venous Access Closure, by Gary X V Tan et al.

- Gelfoam for Closure of Large Percutaneous Transhepatic and Transsplenic Puncture Tracts in Pediatric Patients by Uller et al.

17. Line 320 – 322: “In tract embolization with conventional embolic materials using gelatin sponge, whether the sponge was completely retained within the tract remained uncertain.”

This sentence is incomprehensible in its current form.

18. Lines 322 – 324: “Clinicians have voiced concerns about the displacement of conventional materials from the tract, which might result in tract bleeding.”

Does this statement correspond to the internal experiences of the authors or is it based on the reports from the literature? If the latter applies, please cite the corresponding literature.

19. Do the authors also have experience with the embolization of transhepatic biliary tract routes utilizing the gel-like embolic agent?

20. Line 371 – 372: I don´t really see the point in mentioning in the limitation section that embolization in the surgically exposed animal kidneys was faster than in clinical routine. You might mention this in the results section if appropriate.

6. PLOS authors have the option to publish the peer review history of their article (what does this mean?). If published, this will include your full peer review and any attached files.

Reviewer #1: No

---

## [Author Response · Author response to Decision Letter 0]

16 Oct 2022

Thank you for giving us this opportunity to submit a revised version of our manuscript. We appreciate the comments from the reviewers, which have enabled us to greatly improve our manuscript. We have modified the manuscript based on the helpful comments provided. We have added the requested material to the revised version of the manuscript as concisely as possible. We believe that the revised manuscript will be of interest to the readers of PLOS One.　We would appreciate it if you could review the attached file, "Response To Reviewers" for detailed responses to each comment.

---

## [Decision Letter · Decision Letter 1]

7 Dec 2022

PONE-D-22-16655R1Feasibility of gel-like radiopaque embolic material using gelatin sponge and contrast agent for tract embolization after percutaneous treatmentPLOS ONE

Dear Dr. Kageyama,

Thank you for submitting your manuscript to PLOS ONE. After careful consideration, we feel that it has merit but does not fully meet PLOS ONE’s publication criteria as it currently stands. Therefore, we invite you to submit a revised version of the manuscript that addresses the points raised during the review process.

We look forward to receiving your revised manuscript.

Kind regards,

Peter R. Corridon

Academic Editor

PLOS ONE

Journal Requirements:

Reviewers' comments:

Reviewer's Responses to Questions

**Comments to the Author**

1. If the authors have adequately addressed your comments raised in a previous round of review and you feel that this manuscript is now acceptable for publication, you may indicate that here to bypass the “Comments to the Author” section, enter your conflict of interest statement in the “Confidential to Editor” section, and submit your "Accept" recommendation.

Reviewer #1: All comments have been addressed

Reviewer #2: All comments have been addressed

2. Is the manuscript technically sound, and do the data support the conclusions?

Reviewer #1: Yes

Reviewer #2: Yes

3. Has the statistical analysis been performed appropriately and rigorously? 

Reviewer #1: Yes

Reviewer #2: Yes

4. Have the authors made all data underlying the findings in their manuscript fully available?

Reviewer #1: Yes

Reviewer #2: Yes

5. Is the manuscript presented in an intelligible fashion and written in standard English?

Reviewer #1: Yes

Reviewer #2: Yes

6. Review Comments to the Author

Reviewer #1: Thank you for revising the manuscript and adressing the comments. I do believe that the paper is now worthy of publication.

Reviewer #2: Manuscript Number: PONE-D-22-16655R1

Full Title: Feasibility of gel-like radiopaque embolic material using gelatin sponge and contrast

agent for tract embolization after percutaneous treatment

Declaring competing interests: The reviewer declares that there are no competing interests.

Manuscript summary: This study showed material viscosity test, animal test, and clinical study for appropriate embolic material after percutaneous access and suggested mixture of 3 ml of contrast media and 2 sheets of Gelfoam could be an efficient and safe materials, and it has economic benefit.

Scientific comments: (pages and lines are based on annotated manuscript)

1. Page2 Line28: ‘rabbit kidney’ is not clear, specified it such as ‘rabbit kidney puncture site’.

2. P3L52: ‘intensive surgical approaches’: invasive or extensive would be appropriate than intensive.

3. P7 table 1. Age: give mean and SD, if it is not parametric, give median and ranges.

4. Table 1. PTPE should be more specified, please give numbers of ipsilateral or contralateral access.

5. Table 1. Portal branch for percutaneous access route

Does Right mean right main portal vein? Please more specifically describe.

6. Table 1. Title ‘Postoperative hemorrhage and portal vein thrombosis’ as adverse events.

7. P9L118 ‘All these squares’ would be better than ‘These squares’.

8. P10L148 In response to this line, give mean day and SD of CT obtain in results section.

9. P19 Fig 3. The phases “A 7-Fr sheath (black arrowheads) …” are unnecessarily repeated.

10. P21L329 It would be better to note the difference between reference 21 and this study.

11. The actual videos were not matched with S1 and S2 video legends. S2 is making video and S1 is procedure.

Summary: Interesting topic. Some revisions can improve readability.

7. PLOS authors have the option to publish the peer review history of their article (what does this mean?). If published, this will include your full peer review and any attached files.

Reviewer #1: **Yes: **Anne Marie Augustin

Reviewer #2: **Yes: **DONG JAE SHIM

---

## [Author Response · Author response to Decision Letter 1]

20 Jan 2023

Reviewer #1: 

We appreciate your positive comments that our revised version of the manuscript is worthy of publication. We are grateful for the time and energy you expended.

Reviewer #2:

Thank you for your kind consideration of our manuscript. We are grateful for your comments, which helped us to improve the manuscript. Please find below our point-by-point responses to the comments provided.

---

## [Editor Report · Decision Letter 2]

23 Jan 2023

Feasibility of gel-like radiopaque embolic material using gelatin sponge and contrast agent for tract embolization after percutaneous treatment

PONE-D-22-16655R2

Dear Dr. Kageyama, 

We’re pleased to inform you that your manuscript has been judged scientifically suitable for publication and will be formally accepted for publication once it meets all outstanding technical requirements.

Kind regards,

Peter R. Corridon

Academic Editor

PLOS ONE

---

## [Editor Report · Acceptance letter]

26 Jan 2023

PONE-D-22-16655R2 

Feasibility of gel-like radiopaque embolic material using gelatin sponge and contrast agent for tract embolization after percutaneous treatment 

Dear Dr. Kageyama:

I'm pleased to inform you that your manuscript has been deemed suitable for publication in PLOS ONE. Congratulations! Your manuscript is now with our production department. 

Kind regards, 

on behalf of

Dr. Peter R. Corridon 

Academic Editor

PLOS ONE